# A Delphi technique toward the development of a cognitive intervention framework module for breast cancer survivors with cognitive impairment following chemotherapy

**Syarifah Maisarah Syed Alwi[1], Mazlina Mazlan[2], Nur Aishah Mohd Taib[1], Normah Che Din[3], Vairavan Narayanan[1] ***

**1** Department of Surgery, Faculty of Medicine, University Malaya, Kuala Lumpur, Malaysia, **2** Department of Rehabilitation Medicine, Faculty of Medicine, University Malaya, Kuala Lumpur, Malaysia, **3** School of Healthcare Sciences, Faculty of Health Science, Universiti Kebangsaan Malaysia, Kuala Lumpur, Malaysia

* nvairavan@um.edu.my

**Data Availability Statement:** All relevant data are available from the figshare database: https://doi.org/10.6084/m9.figshare.20024276.v1.

## Abstract

### Objective

Chemotherapy-related cognitive impairment (CRCI) is a well-known phenomenon among breast cancer survivors. Cognitive impairment among breast cancer survivors can significantly affect their quality of life and ability to function independently. However, there is a lack of specific and focused cognitive intervention to improve their cognitive performances. This study aimed to develop a tailored cognitive intervention framework module by adapting the attention and memory interventions from the Cognitive Rehabilitation Manual of the Brain Injury Interdisciplinary Special Interest Group (BI-SIG) of the American Congress of Rehabilitation Medicine (ACRM) and incorporating them with the relevant exercises for cognitive rehabilitation for Malaysian breast cancer survivors with CRCI based on the consensus agreement of the expert panel.

### Methods and analysis

The Delphi consensus technique was conducted online to review and evaluate the framework module. A panel of experts, including rehabilitation medicine physicians, occupational therapists, and clinical psychologists in Malaysia, was invited to participate in this study. For each round, the expert consensus was defined as more than 90% of the expert panel agreeing or strongly agreeing with the proposed items.

### Results

A total of 33 practitioners completed the three Delphi rounds. 72.7% of the expert panel have been practising in their relevant clinical fields for more than six years (M = 10.67, SD = 5.68). In Round 1, 23% of the experts suggested that the framework module for attention training required further improvements, specifically in the language (M = 1.97, SD = 0.75) and instructions (M = 2.03, SD = 0.71) provided. In Round 2, 15% of the experts

**Funding:** The author(s) received no specific funding for this work.

**Competing interests:** The authors have declared that no competing interests exist.

recommended additional changes in the instruction (M = 2.15, SD = 0.67) for attention training. Amendments made to the framework module in line with the recommendations provided by the experts resulted in a higher level of consensus, as 94% to 100% of the experts in Round 3 concluded the framework module was suitable and comprehensive for our breast cancer survivors. Following the key results, the objectives were practical, and the proposed approaches, strategies, and techniques for attention and memory training were feasible. The clarity of the instructions, procedures, verbatim transcripts, and timeframe further enhanced the efficacy and utility of the framework module.

## Conclusions

This study found out that the cognitive intervention framework module for breast cancer survivors with cognitive impairment following chemotherapy can be successfully developed and feasible to be implemented using Delphi technique.

## Introduction

Cognitive intervention is a behaviour-oriented intervention tailored to address cognitive impairment through nonpharmacological techniques [1, 2]. Effective cognitive interventions have been successfully developed for patients with traumatic brain injury [3]. In contrast, very few cognitive interventions have been developed for patients with cancer-related cognitive impairment (CRCI), particularly among female breast cancer patients treated with chemotherapy [1, 4]. Although standard chemotherapeutic agents cannot cross the blood-brain barrier (BBB), findings from a majority of studies demonstrated that breast cancer survivors treated with chemotherapy experienced post-treatment cognitive impairment [5]. Therefore, a possible explanation for this phenomenon was attributable to neurotoxic chemotherapeutic agents, which crossed the BBB into the brain parenchyma and disrupted normal cognitive functioning, notably in the domains of memory, attention, and executive function [6, 7].

According to the literature, the prevalence of CRCI among breast cancer survivors ranges between 16% and 50% within six weeks to nine months following chemotherapy, and the condition may persist for up to twenty years [8–11]. The incidence of cognitive impairment ranges between 16% and 48% immediately to twelve months following chemotherapy. The cognitive domains found impaired were attention, learning, processing speed, executive function, memory, and motor skills [12–17]. In our setting, 30.6% of breast cancer survivors demonstrated cognitive impairment one to three years post-chemotherapy [10]. Cognitive impairment may also lead to the inability to function independently and subsequently disrupt the patient's quality of life [3, 18–21]. Hence, cognitive intervention is increasingly gaining an important role in rehabilitating cognitive impairment resulting from the debilitating effects of chemotherapy among breast cancer survivors.

With regard to the provision of cognitive intervention, the Cognitive Rehabilitation Task Force of the Brain Injury Interdisciplinary Special Interest Group (BI-SIG) of the American Congress of Rehabilitation Medicine (ACRM), cognitive intervention encompasses two distinct approaches, namely cognitive training and compensatory strategies [22]. Cognitive training is used to restore cognitive function and protect the cognitive reserve via individual-based or group-based guided, repetitive cognitive exercises to address impairment in memory, attention, or other cognitive functions [2]. Compensatory strategies, on the other hand, refer to

learning new methods to perform and complete a task or function independently in the presence of cognitive impairment [22–26].

Previously published systematic reviews of cognitive intervention that evaluated the effects of cognitive rehabilitation programs on cognitive impairment among participants with non-central nervous system (CNS) cancer patients have produced inconsistent findings [27, 28]. For example, despite an inadequate number of high-quality clinical trials, Fernandes and colleagues [27] made an explicit recommendation on the effectiveness of the cognitive rehabilitation programs comprising computer training (CT) and strategy training (ST) among mixed cancer survivors. Contrastingly, Treanor and colleagues did not make a firm recommendation on the effectiveness of non-pharmacological interventions such as cognitive training, compensatory strategy, meditation, and physical activity in improving cognitive impairment among breast cancer survivors [28].

We published a systematic review that used the cognitive intervention approaches for cognitive impairment advocated by the BI-SIG of the ACRM among breast cancer survivors [29]. Our review demonstrated that under the current breast cancer management guidelines, only ten studies evaluated the effectiveness of cognitive intervention comprising cognitive intervention and compensatory strategy for breast cancer survivors to improve their cognitive functioning, a vital element in ensuring the resumption of independent daily functions and activities that safeguard their overall quality of life [23, 29, 30]. Cognitive training programs such as the Advanced Cognitive Training for Independent and Vital Elderly (ACTIVE) and Insight program (Posit Science) improved both immediate and delayed memories and processing speed at a 2-month follow-up with a small effect size (Cohen's *d*) [31, 32]. Compensatory strategies such as attention training, memory training, executive training, Memory and Attention Adaptation Training (MAAT), and Promoting Cognitive Health Program (PCHP) compensated impairment in attention, executive function, memory, learning, and processing speed at post-intervention and a 6-month follow-up with small to large effect sizes (Cohen's *d*) [2, 23, 31, 33–36].

Despite inconsistencies in the findings across studies due to the diversity in methodological approaches, attention and memory remained the most affected cognitive domains following the toxicity of chemotherapy among middle-aged and elderly breast cancer survivors [37, 38]. Unfortunately, there is a lack of standard rigorous methods in developing and evaluating cognitive intervention customised for breast cancer survivors with impairment in attention and memory, thus impeding the administration and implementation of cognitive intervention into routine clinical practice. Furthermore, limited evidence is available on the effectiveness of the cognitive intervention. More treatment options are required to explore better alternatives in ameliorating and compensating impairment in both cognitive domains for breast cancer survivors.

Therefore, in this present study, the proposed attention and memory intervention in the existing cognitive rehabilitation manual by the BI-SIG of the ACRM and exercises from Brain Injury Workbook Exercises for Cognitive Rehabilitation by Powell were utilised because these recommendations were advocated by multiple interdisciplinary groups of clinician-scientists with expertise in mild to chronic cognitive impairment [22, 39, 40]. This study aimed to obtain a consensus agreement from an expert panel on a proposed cognitive intervention framework module by adapting the attention and memory intervention for cognitive rehabilitation and incorporating them with the relevant exercises for cognitive rehabilitation for Malaysian breast cancer survivors with CRCI. In principle, a three-round Delphi method used in this study forms an expert panel, asks questions, synthesises, appraises, communicates feedback, and directs the identified expert panel to consensus building [41–43]. Although the validity of the results is not rooted in statistical significance, multiple opinions obtained from the experts were noteworthy in improving the developed framework module.

## Material and methods

### Study design

This study utilised the Delphi technique conducted between November 2020 and August 2021. The technique comprised three rounds of an email-based survey in reviewing and evaluating the framework module tailored for breast cancer survivors with CRCI. The anonymity in providing opinions was asserted throughout the process, and the experts were allowed to change their earlier judgment in each questionnaire iteration. For each round, controlled feedback was also provided whereby the experts were informed of the opinions of other anonymous experts in the group. The final appraisal of the experts was demonstrated through the statistical average [44, 45]. Approval was granted by the Medical Research Ethics Committee, University Malaya Medical Centre (UMMC) (MREC ID Number: 20201229–9634). Fig 1 illustrates the process of the Delphi study.

### Participants

The expert panel included rehabilitation medicine physicians, occupational therapists, and clinical psychologists. They were selected to provide their opinions and judgments based on their relevant expertise, knowledge, and experiences in cognitive intervention. This study utilised two primary guidelines in forming the expert panel: 1) defining the area of expertise of the potential expert panel and 2) identifying the potential expert panel with relevant expertise.

To classify experts, practitioners must have the pertinent technical knowledge and clinical experience in administering cognitive intervention among patients with cognitive impairment. As prominent experts, they must be proficient in imparting their views, particularly approaches, strategies, or techniques that can be used to improve cognitive impairment and the procedures involved in administering these interventions to the patients.

Actor types and snowball sampling were used to identify the potential expert panel (Table 1). For actor types, potential experts from various affiliations were identified from multiple existing government databases. For snowball sampling, some of the recognised experts from the actor type recommended other experts to participate in this study. The recommended experts were also included in the panel if they fulfilled the inclusion criteria.

The following inclusion criteria were used: 1) rehabilitation medicine physician with at least three years of clinical experience in cognitive rehabilitation, 2) occupational therapist with at least three years of clinical experience in cognitive rehabilitation, 3) clinical psychologist with at least three years of clinical experience in cognitive rehabilitation, and 4) practising in Malaysia.

### Data collection procedures

The expert panel was required to spend 90 minutes, spread over three rounds (Round 1 to Round 3) of an email-based survey (Fig 1). A survey coordinator was appointed to manage the entire process of the Delphi study. The survey coordinator was responsible to identify and recruit the expert panel and send the formal invitation, reminder, and acknowledgment emails. The expert panel was contacted through institutional, corporate, or organisational email addresses to prevent information leakage and eliminate spam emails [46].

Three weeks before the start of Round 1, the survey coordinator contacted and invited a prospective expert panel to participate in this study via email. A brief description of the goals, procedures, number of questionnaires to be completed, and expected timeframe of the overall study was provided in the email. Those who agreed to participate were required to complete an electronic form containing their name, area of expertise, updated place of clinical practice,

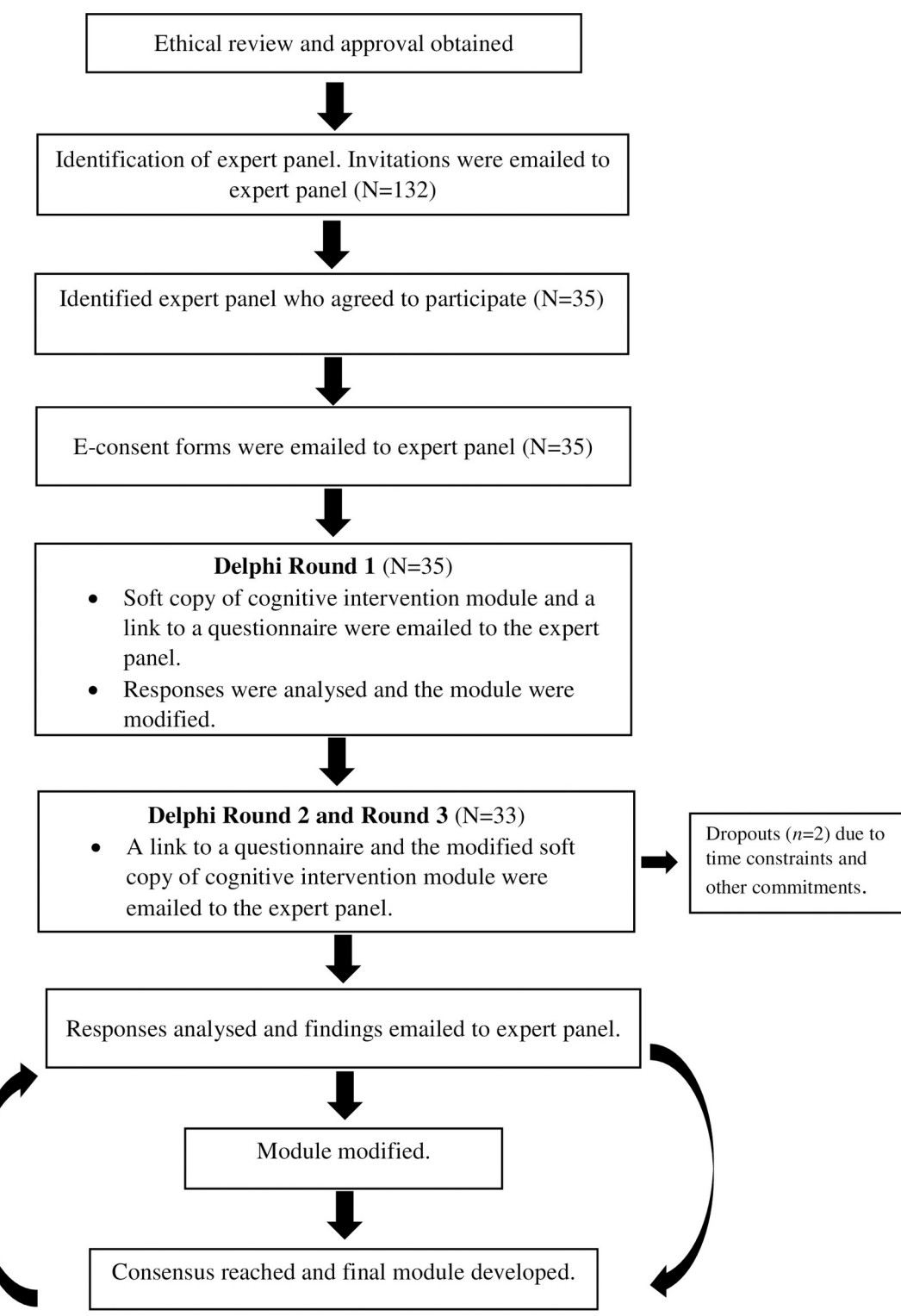

**Fig 1. The flow of the Delphi study.**

The process of the Delphi study presented comprised three rounds.

Fig 1. The flow of the Delphi study. The process of the Delphi study presented comprised three rounds.

**Table 1. Matrix of identifying expert panel.**

| Expert panel profession | Actor types | | | Snowball sampling |
|---|---|---|---|---|
| | NSR | MOTA | MSCP | (*n*) |
| | (*n*) | (*n*) | (*n*) | |
| Rehabilitation medicine physician | 36 | - | - | 3 |
| Occupational therapist | - | 27 | - | 6 |
| Clinical psychologist | - | - | 58 | 2 |

Remarks: NSR- National Specialist Register of Malaysia, MOTA- Malaysian Occupational Therapy Association, MSCP- Malaysian Society of Clinical Psychology.

and years of clinical practice. Those who declined to participate were removed from the subsequent follow-up email. In the follow-up email, the expert panel was provided with a link to the electronic documentation of informed consent or e-consent. Upon obtaining their consent, they would be considered as part of the expert panel of the Delphi study.

In Round 1, the expert panel was given an email containing a detailed description of the Delphi procedure and a soft copy of the framework module. Next, they were instructed to review and evaluate the framework module by completing the online questionnaire attached within two to four weeks. A reminder to complete the first round of questionnaires was emailed to the experts three days before the deadline.

Feedback from the experts in Round 1 was collated and analysed to improve the framework module before formulating the questionnaire for the following rounds. In Round 2 and Round 3, the expert panel was required to rate the importance of all feedback provided by the experts to achieve a final consensus on the developed framework module. The same procedures from Round 1 were repeated. However, in Round 2, a summary of feedback from Round 1 and a revised framework module were presented to the experts. For Round 3, a summary of responses from Round 2 and a revised framework module were disseminated to the experts.

## Instruments

For Round 1, the questionnaire comprised two sections. The first section contained 57 questions (S1 Table) on a 4-point Likert scale ranging from "3 = strongly agree" to "0 = strongly disagree". In this section, the expert panel evaluated the practicality and comprehensiveness of the objectives, rationales, instructions, procedures, examples, and language used in the framework module. By rating the questions in this section, the expert panel would appraise the extent of the insight among breast cancer survivors in improving their cognitive functioning and compliance with the cognitive intervention.

The second section consisted of ten open-ended questions (S2 Table) to enhance the input from the experts. Responses from the experts were used to provide insights on other potential approaches, strategies, or techniques that can be utilised to improve impairment in the cognitive domains of attention and memory. Questions on the likelihood of the framework module being incorporated into clinical practice and the potential experts responsible for delivering the intervention were also included. Finally, the expert panel was asked about possible benefits, risks, burdens, and ethical issues in implementing this intervention. The responses attained from both sections were used to inform the following round of the survey.

In Round 2, the expert panel was requested to re-rank all 23 items from "3 = strongly agree" to "0 = strongly disagree" using a 4-point Likert scale. The items included in the Round 2 questionnaire comprised 15 new items for the experts to rate after the modification was made to the framework module. The new items were also based on the suggestion of the experts to add

**Table 2. Examples of items in the questionnaire.**

| Component | Example of the question asked |
|---|---|
| Practicality | Are the objectives practical to accomplish? |
| Applicability | Are the activities suggested appropriate? |
| | Can Malaysian breast cancer survivors complete these activities? |
| Feasibility | Are Malaysian therapists capable of delivering this cognitive intervention module? |
| | Who is the best person to deliver this cognitive intervention module? |
| Effectiveness | When should the cognitive intervention module be administered to ensure maximum effectiveness? |
| | How can we monitor the effectiveness of the cognitive intervention module? |
| Utility | What are the benefits if the cognitive intervention module is implemented? |
| Barriers/ Challenges | What are the risks and burdens if the cognitive intervention module is implemented? |

specific verbatim transcripts and timeframes to complete each approach, technique, and activity of the cognitive intervention. Besides, eight items from the Round 1 questionnaire (S3 Table) were retained in this round because less than 10% of the experts disagreed or strongly disagreed with the items. Likewise, in Round 3, the expert panel was instructed to complete the final questionnaire comprising 23 items before a good framework module consensus was achieved. A systematic review conducted by Boulkedid and colleagues suggested that at least 75% of the expert panel must agree to the items in the questionnaire before reaching any consensus on the Delphi technique [47]. In this study, a consensus was deemed to be accomplished if more than 90% of the expert panel rated the items from the questionnaire for each round as "agree" or "strongly agree." Table 2 outlines the examples of items included in the questionnaire.

## Statistical analysis

All statistical analyses were performed using the IBM Statistical Package for the Social Sciences (SPSS) Version 22. Descriptive statistics were used to describe the characteristics of the expert panel. From Round 1 to Round 3, average points or mean (M) and the standard deviation (SD) for each variable were computed. The percentage of the expert panel who rated the items as "agree" and "strongly agree" was also calculated.

For the open-ended questions from Round 1, responses obtained were qualitatively analysed and manually coded. No supporting software was utilised in the data analysis. All authors were delegated specific tasks to complete the qualitative data analysis. First, two authors were independently responsible for the initial manual coding of the major themes from the responses, and any overlapping responses were removed. Next, two other authors independently coded large samples of the responses that were randomly selected. The primary goal of conducting four independent analyses was to attain a consistent conclusion and minimise the unintentional exclusion of any responses the panel gave. While triangulating the panel's responses, any minor differences were resolved through discussion and consensus among all authors [44]. Finally, one author cross-checked each response with the themes developed from the initial coding to establish a good fit [48].

## Results

The characteristics of the expert panel are presented in Table 3. Only 33 experts completed all three rounds of the Delphi study. The experts who participated in the present study were practitioners from multiple practice fields, including rehabilitation medicine, occupational therapy,

**Table 3. Characteristics of the expert panel.**

| Characteristics | *n* | Percentage (%) | M |
|---|---|---|---|
| | (N = 33) | | (SD) |
| Expert panel profession | | | |
| Rehabilitation medicine physician | 6 | 18.2 | |
| Occupational therapist | 14 | 42.4 | |
| Clinical psychologist | 13 | 39.4 | |
| Years of clinical practice | | | 10.67 (5.68) |
| 3 to 5 years | 9 | 27.3 | |
| 6 to 10 years | 10 | 30.3 | |
| 11 to 15 years | 7 | 21.2 | |
| 16 years and above | 7 | 21.2 | |

and clinical psychology. Overall, 72.7% (*n* = 24) of the expert panel members have been practising in their relevant clinical fields for more than six years (M = 10.67, SD = 5.68).

In Round 1 of the Delphi study, the expert panel (N = 35) suggested that both attention and memory training objectives were practical to be accomplished by the breast cancer survivors (S1 and S2 Tables). Almost all (94% to 100%) agreed that the approaches, techniques, and activities introduced in the framework module echoed the training objectives. As for the extent to which the intervention was seen as applicable, 91% to 100% of the expert panel predicted that 83% to 97% of the breast cancer survivors could complete the training. Regarding feasibility, 86% to 97% of them felt that Malaysian therapists were capable of administering the intervention, with 43% mentioning that occupational therapists and clinical psychologists should be responsible for rehabilitating cognitive impairment. Based on the feedback, the likelihood of the framework module being incorporated into clinical practice was very high (89%). Although 60% to 75% of the expert panel responded favourably in terms of the utility of the intervention, more than half (60%) of them proposed that therapists and breast cancer survivors might experience multiple challenges, such as limited transportation and time constraints to attend the intervention session.

In Round 2 and Round 3 (Table 4), some items from Round 1 were retained for further evaluation because the percentage of consensus among the experts was lower than the defined criteria. Additional items were included in the questionnaire after the amendment of the original framework module. Higher levels of consensus were noted in Round 2 as 85% to 100% of the expert panel rated most of the items in Round 2 as "agree" or "strongly agree". In Round 3, 94% to 100% of the expert panel consented that the improvement made on the final version of the framework module was adequately tailored to breast cancer survivors with CRCI.

## Discussion

This study utilised the Delphi method to explore and acquire the opinions of members from an expert panel in cognitive intervention before administering this framework module to our cohort of breast cancer survivors. In Round 1, 35 experts reviewed and evaluated the contents of the framework module. Although 77% to 100% of the experts advocated the comprehensiveness of the framework module, 77% of the experts felt that the suitability of the language and clarity of the instructions provided were unclear and inadequate. Hence, these recommendations were applied in modifying multiple technical issues, such as the practicality of the objectives conveyed for attention and memory training and the applicability and clarity of the instructions, procedures, examples, and timeframe used for each approach, technique, and activity designed. Finally, to monitor the effectiveness of the training in improving attention

**Table 4. Results from Delphi Round 2 and Round 3.**

| Items | Round 2 (N = 33) | | Round 3 (N = 33) | |
|---|---|---|---|---|
| | M | % | M | % |
| | (SD) | Rated | (SD) | Rated |
| | | 2–3 | | 2–3 |
| *Introduction* | | | | |
| 1. Are the descriptions of the cognitive impairment stated clear and sensible? | 2.27 (0.52) | 97 | | |
| 2. Are the descriptions of the content of the cognitive intervention module comprehensive? | 2.27 (0.52) | 97 | | |
| 3. Are the suggestions on whom is appropriate to use the cognitive intervention module understandable? | 2.39 (0.56) | 97 | | |
| 4. Are the explanations on how to use the cognitive intervention module easy to understand? | 2.30 (0.64) | 91 | 2.45 (0.51) | 100 |
| *Objectives of attention training* | | | | |
| 1. Are the objectives stated clearer? | 2.42 (0.50) | 100 | | |
| 2. Are the objectives more practical to accomplish? | 2.33 (0.54) | 97 | | |
| *Definition of attention* | | | | |
| 1. Are the definitions provided precise and easy to understand? | 2.30 (0.47) | 100 | | |
| 2. Are the descriptions and examples of how attention works rendered a better understanding of the definition of attention? | 2.30 (0.53) | 97 | | |
| 3. Are the descriptions and examples stated in the hierarchical components of attention clearer and sensible? | 2.39 (0.61) | 94 | | |
| *Activities for attention* | | | | |
| 1. Are the instructions provided easy to understand? | 2.15 (0.67) | 85 | 2.42 (0.61) | 94 |
| 2. Are the examples provided to do the activities understandable? | 2.18 (0.58) | 91 | 2.36 (0.60) | 94 |
| 3. Is the organization of the activities suggested appropriate? | 2.30 (0.59) | 94 | | |
| 4. Is the recommended timeframe suitable to complete the activities? | 2.18 (0.53) | 94 | | |
| *Approaches to the rehabilitation of attention: N-Back task* | | | | |
| 1. Are the procedures suggested easy to understand? | 2.15 (0.57) | 94 | | |
| *Approaches to the rehabilitation of attention: Time Pressure Management (TPM)* | | | | |
| 1. Are the procedures suggested easy to understand? | 2.24 (0.50) | 97 | | |
| 2. Are the examples provided with a better understanding of the approach? | 2.36 (0.49) | 100 | | |
| *Objectives of memory training* | | | | |
| 1. Are the objectives stated clear and sensible? | 2.36 (0.55) | 97 | | |
| 2. Are the objectives practical to accomplish? | 2.30 (0.53) | 97 | | |
| *Definition of memory* | | | | |
| 1. Is the organization of the subdivisions of memory appropriate? | 2.39 (0.56) | 91 | 2.39 (0.50) | 100 |
| *Approaches to the rehabilitation of memory: Memory Strategy Training- Association techniques and organisational and elaboration techniques* | | | | |

*(Continued)*

**Table 4.** (Continued)

| Items | Round 2 (N = 33) | | Round 3 (N = 33) | |
|---|---|---|---|---|
| | M | % | M | % |
| | (SD) | Rated | (SD) | Rated |
| | | 2–3 | | 2–3 |
| 1. Are the instructions stated clear and sensible? | 2.21 (0.49) | 91 | 2.33 (0.48) | 100 |
| 2. Are the stories under the PQRST strategy suitable? | 2.24 (0.50) | 97 | | |
| 3. Is the organization of the activities suggested appropriate? | 2.27 (0.45) | 100 | | |
| 4. Is the recommended timeframe suitable to complete the activities? | 2.15 (0.51) | 94 | 2.18 (0.47) | 97 |

Remarks: 3 = Strongly agree, 2 = Agree.

and memory, we added supplementary materials comprising a list of recommended neuropsychological test batteries and a self-reported cognitive performance questionnaire in the module.

Throughout Rounds 2 and 3, 33 experts evaluated the modified version of the framework module before reaching a consensus. In the second round, as high as 85% to 100% of experts rated the framework module as covering all important aspects of improving attention and memory. To further enhance the feasibility and applicability of the framework module, an introduction section was added based on the responses from Round 1. This section provided an in-depth description of cognitive impairment as additional input for prospective therapists to enhance their awareness of cognitive impairment for breast cancer survivors. Based on further recommendations in Round 1, a comprehensive description of the contents, suggestions on the potential experts responsible for administering the cognitive training, and a step-by-step guideline on using the framework module were also added in the introduction section. In addition, specific verbatim transcripts for each approach, technique, and activity were provided to ensure that potential therapists abide by standard procedures when administering this framework module.

Apart from that, considering this framework module should be culturally appropriate for Malaysian breast cancer survivors, we carefully selected the appropriate names, fruits, food, places, and stories to be used based on the experts' feedback. Upon completing the amendment, 94% to 100% of the expert panel determined the framework module would be comprehensive for Malaysian breast cancer survivors. Furthermore, the expert panel concluded that the framework module could be incorporated into clinical practice for cognitive rehabilitation. Malaysian therapists such as occupational therapists and clinical psychologists were identified as the experts capable of delivering the intervention. Although the framework module provided a standard protocol, guidelines, and verbatim transcripts, the expert panel recommended that each potential expert undergo appropriate training before initiating the intervention to the breast cancer survivors to ensure optimal effectiveness.

During the Delphi process, multifaceted burdens, including time constraints and increased workload, were recognised as possible challenges therapists face in administering the intervention. Regardless of the challenges, the expert panel acknowledged that both the attention and memory training in the framework module should improve the breast cancer survivors' psychological well-being and ability to carry out their daily activities independently. According to the Malaysia National Cancer Registry Report, from 2012 to 2016, 41.9% of women diagnosed

with breast cancer were aged 25 to 59 [49]. At the same time, 38.9% of the Malaysian labour force consisted of females [50]. Many of the Malaysian breast cancer survivors were younger and working adults. Hence, cognitive impairment may place them at a disadvantage in the workplace. As a result, the expert panel deemed that the training in this framework module would be vital in improving their work performance and work productivity.

Although the time gap between the completion of chemotherapy and the implementation of the cognitive intervention framework module remains unclear, the expert panel felt that the cognitive intervention should ideally be initiated among breast cancer survivors who have completed their chemotherapy. This is appropriate because research has shown that spontaneous recovery of cognitive function is more likely to occur immediately after chemotherapy [1, 2, 51]. In addition, the effectiveness of the cognitive intervention can be determined via the signs of cognitive improvement following the administered intervention. The expert panel also recommended proper monitoring of the effectiveness of the cognitive intervention module via three time-points of the cognitive assessments at baseline, during, and post-intervention.

In summary, the three-round Delphi approach successfully obtained and assessed opinions from the expert panel to guide the development of a culturally appropriate framework module for our breast cancer survivors from multiethnic and multiracial backgrounds. The expert panel concluded that the framework module would benefit breast cancer survivors with cognitive impairment. We acknowledged that the administration method and the efficacy of the framework module should be examined further with a pilot test and, subsequently, a randomised clinical trial among breast cancer survivors to assess the clinical effectiveness of the cognitive training framework module.

## Critical reflection on the study and the study results

To the best of our knowledge, this is the first study that uses a Delphi technique in cognitive rehabilitation. The findings obtained from the present study were based on the knowledge, opinions, and practical experiences of the expert panel in cognitive intervention rather than empirical evidence. The results obtained from this Delphi process might be subjected to researcher bias. For example, there was no direct interaction between the expert panel and the survey coordinator in developing the questionnaire by the research group, which could lead to subjective interpretations of responses obtained by the research group. In view of this, the findings from this study should be interpreted cautiously.

Although the nature of the Delphi method is highly selective about the types of participants, we acknowledge that it is unlikely that all Malaysian experts in cognitive rehabilitation participated in this study. The three databases used to identify and recruit the expert panel might be incomplete. The data provided might not be up to date, thus rendering the likelihood of missing out on potential experts reviewing and evaluating the framework module.

There are currently no standard criteria available in the existing literature to define an acceptable consensus within the expert panel. Our study set higher criteria of 90% for consensus than other studies with similar purposes [52, 53]. Although we successfully obtained a complete consensus on the comprehensiveness of the framework module for our cohort of breast cancer survivors, we agreed that the framework module should be examined for its effectiveness among our cohort of breast cancer survivors.

The strength of our study resides in the heterogeneity of the expert panel backgrounds and clinical experiences. By using snowball sampling, we could control the selection bias that may have occurred in this study. In contrast to a focus group, the Delphi study does not require proximity and emphasises a structured anonymous communication of opinions and responses [41, 43, 54]. Through online questionnaires, all experts could freely express their responses

and judgements on the framework module confidentially without being influenced by other dominant experts, which helped reduce the risk of group dynamics to negative influence conclusions [43]. Hence, the responses collected could be aggregated and evaluated based on their true merits.

Unlike other group methods, such as a focus group, the flexibility and reflexivity of the Delphi method allow us to accommodate the techniques to our research context [43, 54]. For example, questionnaires used for the data collection enrich our insight into the comprehensiveness of the framework module. The multiple iterations through three Delphi rounds enabled and encouraged the expert panel to reassess and reconsider their judgements. Indeed, through the Delphi method, the research team was able to incorporate valuable feedback by refining the contents of the framework module in each round, thus, enhancing the validity of the outcomes of our study. The statistical aggregation of the group responses served as the controlled feedback that reduced conflicts because the experts were presented with anonymous opinions obtained from other experts in the same Delphi study [44, 45]. Finally, this study is replicable in each setting as the methodology and procedures have been clearly outlined.

## Conclusion

The long-term adverse effects of chemotherapy on cognitive impairment among breast cancer survivors can significantly hamper their cognitive performances and multidimensional quality of life. Therefore, it is crucial to incorporate customised cognitive rehabilitation for this cancer population into routine clinical practice. The three-round Delphi approach successfully obtained and assessed opinions from the expert panel to guide the development of an appropriate cognitive intervention framework module for Malaysian breast cancer survivors with impairment in attention and memory. In the final round of the Delphi process, the expert panel concluded that the objectives, approaches, techniques, and exercises proposed in the framework module are practical, applicable, feasible, and would be advantageous for Malaysian breast cancer survivors. We acknowledged that the contents of the framework module are culturally appropriate for Malaysian breast cancer survivors from multiethnic and multiracial backgrounds. For example, the names, fruits, and food used in the framework module represent the diversity among ethnicity and races in Malaysia. Additionally, this framework module can be applied to other Southeast Asia countries, specifically Singapore, Indonesia, Brunei, and Thailand. However, the administration method and the efficacy of the framework module should be examined further with a pilot test and, subsequently, a randomised clinical trial among breast cancer survivors to assess the clinical effectiveness of the framework module.

## Supporting information

**S1 Table. Results from Delphi Round 1 (N = 35).**
(PDF)

**S2 Table. Major themes from coding the responses obtained in Round 1 (N = 35).**
(PDF)

**S3 Table. Items retained from the Round 1 questionnaire.**
(PDF)

## Acknowledgments

All authors would like to acknowledge and express our gratitude to all the study participants.

## Author Contributions

**Conceptualization:** Syarifah Maisarah Syed Alwi, Mazlina Mazlan, Vairavan Narayanan.

**Data curation:** Syarifah Maisarah Syed Alwi, Vairavan Narayanan.

**Formal analysis:** Syarifah Maisarah Syed Alwi, Nur Aishah Mohd Taib, Normah Che Din, Vairavan Narayanan.

**Methodology:** Syarifah Maisarah Syed Alwi, Mazlina Mazlan, Vairavan Narayanan.

**Project administration:** Syarifah Maisarah Syed Alwi, Mazlina Mazlan, Vairavan Narayanan.

**Writing – original draft:** Syarifah Maisarah Syed Alwi, Mazlina Mazlan, Nur Aishah Mohd Taib, Normah Che Din, Vairavan Narayanan.

**Writing – review & editing:** Syarifah Maisarah Syed Alwi, Mazlina Mazlan, Nur Aishah Mohd Taib, Normah Che Din, Vairavan Narayanan.

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
